# Genome-Wide Association Studies and Transcriptome Changes during Acclimation and Deacclimation in Divergent *Brassica napus* Varieties

**DOI:** 10.3390/ijms21239148

**Published:** 2020-11-30

**Authors:** David P. Horvath, Jiaping Zhang, Wun S. Chao, Ashok Mandal, Mukhlesur Rahman, James V. Anderson

**Affiliations:** 1USDA-ARS, Sunflower and Plant Biology Research Unit, Edward T. Schafer Agricultural Research Center, 1616 Albrecht Blvd., N., Fargo, ND 58102-2765 1, USA; Wun.Chao@usda.gov (W.S.C.); James.V.Anderson@usda.gov (J.V.A.); 2Physiology and Molecular Biology Laboratory of Ornamental Plants, Institute of Landscape Architecture, College of Agriculture & Biotechnology, Zhejiang University, Hangzhou 310058, Zhejiang Province, China; zhangjiaping0604@aliyun.com; 3Department of Plant Sciences, North Dakota State University, Fargo, ND 58104 3, USA; Ashok.Mandal@ndsu.edu (A.M.); md.m.rahman@ndsu.edu (M.R.)

**Keywords:** freezing tolerance, deacclimation, GWAS, transcriptomics, enrichment analysis, promoter motifs

## Abstract

Information concerning genes and signals regulating cold acclimation processes in plants is abundant; however, less is known about genes and signals regulating the deacclimation process. A population of primarily winter *B. napus* varieties was used to conduct a genome-wide association study and to compare the transcriptomes from two winter *B. napus* varieties showing time-dependent differences in response to cold acclimation and deacclimation treatments. These studies helped to identify loci, candidate genes, and signaling processes impacting deacclimation in *B. napus*. GWAS identified polymorphisms at five different loci associated with freezing tolerance following deacclimation. Local linkage decay rates near these polymorphisms identified 38 possible candidate genes. Several of these genes have been reported as differentially regulated by cold stress in arabidopsis (*Arabidopsis thaliana*), including a calcium-binding EF-hand family protein (encoded by BnaCnng10250D) that was also differentially expressed during deacclimation in this study. Thousands of other genes differentially expressed during the acclimation and deacclimation treatments implicated processes involving oxidative stress, photosynthesis, light-regulated diurnal responses, and growth regulation. Generally, responses observed during acclimation were reversed within one week of deacclimation. The primary differences between the two winter *B. napus* varieties with differential deacclimation responses involved protection from oxidative stress and the ability to maintain photosynthesis.

## 1. Introduction

*Brassica napus* is a valuable oilseed crop of the northern Great Plains of the USA and Canada. Winter varieties planted in the fall and that vernalize over the winter have potential as cover crops for weed suppression and are higher yielding and faster flowering than spring varieties [1]. However, winter varieties of *B. napus* rarely survive the extreme over-wintering conditions experienced in the northern Great Plains region, which makes improving freezing tolerance a critical component for the survival of winter biotypes in this growing region. Plants that receive the proper cold treatment and duration have increased ability to survive freezing temperatures, due to a process known as cold acclimation [2]. Although plants like *B. napus* take weeks to properly cold-acclimate, it often only takes a few days at growth-conducive temperatures for some winter *B. napus* plants to lose freezing tolerance, a process often referred to as deacclimation.

Considerable work has gone into understanding how cold acclimation is induced in arabidopsis (*Arabidopsis thaliana* Heyn) and how molecular signals responsible for regulating gene expression and subsequent changes in cell physiology allow plants to survive freezing [3,4]. Studies have identified a core set of COLD-REGULATED (COR) genes and the key transcription factors such as CRT/DRE BINDING FACTOR (CBFs) and ABSCISIC ACID RESPONSIVE ELEMENTs (ABREs) that are responsible for their induction [2], as well as the transcription factors that induce CBF genes in response to cold [3]. However, the receptor of the temperature signal has not been identified—although mutant analyses have implicated intrinsic membrane proteins that act as calcium channels being linked to the processes [5].

Less is known about the signals controlling deacclimation, although several studies have examined transcriptional changes associated with the deacclimation process in both arabidopsis [6] and rapeseed [7]. Both studies implicated CBF signaling to be a significant player in the deacclimation process, with several arabidopsis COR genes [6] and several *B. napus* CBF genes being down-regulated [7] within 24 h following deacclimation. Several studies have also associated deacclimation rates and intensity with loss of proline accumulation/biosynthesis, reactive oxygen species and regulators thereof, and altered sugar levels and composition [6,8], but the regulatory genes controlling these responses have not yet been defined.

Differences in the deacclimation responses of both arabidopsis and *B. napus* varieties have been noted [9,10]. For example, different rates of deacclimation have been observed in response to diurnal light and temperature regimes [11]. Intriguingly, varieties with the greatest freezing tolerance appeared to deacclimate faster than varieties with less freezing tolerance [8,9].

Another way to identify genes involved in physiological and developmental processes is through the use of genome-wide association studies (GWAS) [12]. Such studies use genetic variation present in a population, together with phenotypic analysis, to identify genetic markers linked to phenotypes of interest. By looking at genes that are genetically linked to the associated markers, it is sometimes possible to identify the genes responsible for the phenotypic variation in the population. A limited number of studies have used genetic linkage to identify genes associated with freezing tolerance [13,14], but none have looked at loci associated with deacclimation processes.

This study demonstrates a range of varietal differences in freezing tolerance following deacclimation among a diversity panel of primarily winter *B. napus* varieties. We used an association mapping strategy to identify loci linked to the intensity of deacclimation, and present data showing differences in freezing tolerance following deacclimation under different temperature regimes in two diverse winter *B. napus* varieties. We also present data on the transcriptome changes associated with these two diverse varieties at an intermediate temperature that results in full deacclimation of one variety, but only partial deacclimation of the other variety.

## 2. Results

### 2.1. Genotypic Differences in Response to Deacclimation Temperatures

Phenotypic analysis of a diversity panel composed primarily of 397 winter *B. napus* varieties was conducted to determine the genotypic variation in response to deacclimation within the population. Preliminary studies on one fourth of the population indicated that nearly 100% survive after a four hour freezing regime at −10 °C if first acclimated at 5 °C for four weeks, while 100% of the unacclimated plants did not survive the freezing regime. Further, a three-day deacclimation period reduced but did not completely remove the ability of some lines to survive the freezing regiment. Thus, we subjected the entire population to these conditions, and phenotyped the population for freezing damage (Figure 1 and Appendix A). Considerable variation was observed by genotype with 16 lines completely loosing freezing tolerance, and 13 lines showing minimal loss of freezing tolerance following deacclimation.

Two winter *B. napus* varieties (KS4666 and KS09068B-5-1) showing differential freezing tolerance following deacclimation were chosen for further study. Deacclimation for three days at 15 °C constant temperature resulted in near complete loss of freezing tolerance in both varieties (Figure 2). However, deacclimation for three days at 10 °C constant temperature resulted in freezing damage that was variable between varieties KS4666 and KS09068B-5-1, with KS09068B-5-1 showing damage comparable to fully deacclimated plants and KS4666 showing freezing tolerance similar to acclimated plants (Figure 2). Interestingly, deacclimation at 10 °C for a longer period of time (one–two weeks) did not significantly reduce the freezing tolerance of KS4666 (Figure 3). These results highlight the significant divergence in deacclimation between KS4666 and KS09068B-5-1 following four weeks of cold acclimation.

### 2.2. RNAseq Analysis Identified Differential Expression between Treatments and between Varieties

RNAseq was accomplished using winter *B. napus* leaf tissue collected before (0A0D) and after four weeks of acclimation (4A0D) and after one (4A1D) or two weeks (4A2D) of deacclimation at 10 °C (Appendix A). Depending on the treatment, between 26 K and 30 K genes (those with an FPKM > 5 in all three replicates for any given treatment) were expressed. Differential gene expression (FDR < 0.05 and log fold change >1) was observed for each pairwise comparison between treatments, and between varieties at each treatment. Of the 101,040 gene models mapped, between ~24 K and ~30 K were expressed in one or both treatments being compared. There were 4490 genes differentially expressed between varieties under control (unacclimated) conditions, which represents their varietal differences (Figure 4 and Appendix A orange and green highlighted genes on page “0A KS09068B-5-1 vs. 0A KS4666”). Although both varieties acclimate to similar levels after four weeks at 5 °C, surprisingly, only 2911 genes were differentially expressed between these varieties in the fully acclimated plants (Appendix A orange and green highlighted genes on page “4A KS09068B-5-1 vs. 4A KS4666”). Since 1522 of those 2911 genes were not represented among the genes differentially expressed under control conditions (Appendix A yellow highlighted “gene_id” page “4A KS09068B-5-1 vs. 4A KS4666”), these could represent some varietal differences specific to acclimation. There were 3553 and 4396 genes differentially expressed after one and two weeks deacclimation at 10 °C, respectively, between the two winter *B. napus* varieties (Figure 4 and Appendix A orange and green highlighted cells pages “1D KS09068B-5-1 vs. 1D KS4666” and “2D KS09068B-5-1 vs. 2D KS4666”). A set of 673 genes was differentially expressed between these two varieties under all conditions (Figure 4 and Appendix A pages “up in KS09068B-5-1 vs. KS4666” and “down in KS09068B-5-1 vs. KS4666” final columns).

When comparing treatment effects, variety KS4666 had 10,038 genes differentially expressed following four weeks of cold acclimation as compared to control conditions; meanwhile, variety KS09068B-5-1 had 8942 genes differentially expressed (Figure 5 and Appendix A orange and green highlighted cells on ages “0A KS4666 vs. 4A KS4666” and “0A KS09068B-5-1 vs. 4A KS09068B”). Of these, more than half (5820 genes) were differentially expressed in both varieties and with the same trend in response to the cold acclimating condition, which likely represent a core set of cold-regulated genes in these two winter *B. napus* varieties (Figure 5 and Appendix A, yellow or blue highlighted gene_id cells on page “0A KS4666 vs. 4A KS4666”).

Surprisingly, several COR genes (specifically COR15a and COR15b) were strongly down-regulated in acclimated as compared to unacclimated plants in both varieties (see Discussion for possible reasons). A comparison between fully acclimated and one-week deacclimation indicated that 2222 genes were differentially expressed in variety KS4666, while 2849 were differentially expressed in variety KS09068B-5-1 (Figure 5). Of these, 770 were differentially expressed with the same trend in both varieties (Appendix A, yellow highlighted gene_id cells on page “4A KS4666 vs. 1D KS4666”). Of the 770 genes that were differentially expressed between fully acclimated and one-week deacclimated plants in both varieties, 516 were also differentially expressed between unacclimated and acclimated plants in both varieties but with the opposite trend (Appendix A, blue highlighted gene_id cells on page “0A KS4666 vs. 4A KS4666”). These genes likely represent those that consistently responded to deacclimation by returning to unacclimated states—that is, they are differentially regulated during acclimation, but return to unacclimated levels following deacclimation. Although COR genes were down-regulated in acclimated plants compared to unacclimated plants, their expression was further down-regulated during deacclimation. It is noteworthy that variety KS4666, which had minimal loss of acclimation during the deacclimating process, also showed the fewest genes returning to normal unacclimated levels. However, there were still many differences in gene expression between the unacclimated and deacclimated plants. For example, only 1469 and 985 genes were differential between one- and two-week deacclimated plants for varieties KS4666 and KS09068B-5-1, respectively (Appendix A, orange and green highlighted cells, pages “1D KS4666 vs. 2D KS4666” and “1D KS09068B-5-1 vs. 2D KS09068B”), suggesting most of the deacclimation happens in the first week at 10 °C.

### 2.3. Gene Set Enrichment Analysis Identified Physiological Processes and Regulatory Components Associated with Freezing Tolerance

The 5820 genes differentially expressed during cold acclimation in winter *B. napus* varieties KS4666 and KS09068B-5-1 were used for multiple gene set enrichment analyses (Appendix A, yellow and blue highlighted gene_ids, page “1D KS4666 vs. 2D KS4666”). Among the genes up-regulated by cold acclimation in both varieties, 1660 arabidopsis gene models were represented (Appendix A, yellow highlighted cells in column AGI, page “0A KS4666 vs. 4A KS4666”). Even though the CBF regulon seemed to be repressed in cold-acclimated relative to unacclimated plants, CBF binding sites were still over-represented, indicating a significant subset of genes up-regulated by cold acclimation in these winter *B. napus* varieties were under control of the CBF regulon. However, other notable transcription factor binding sites were also over-represented. These included the RELATIVE OF EARLY FLOWERING 6 (REF6) binding site, and several others for previously characterized ERF/AP2 transcription factors including DREBs 2 and 26 (Figure 6 and Appendix A, page “Motif_enrichment”). Likewise, collections of ChIP studies indicated support for CCA1 binding among many other previously studied transcription factors. Only six KEGG ontologies were significantly over-represented including ribosome, ribosome biogenesis in eukaryotes, circadian rhythm, flavonoid biosynthesis, glucosinolate biosynthesis, and nitrogen metabolism (Appendix A, page “KEGG_enrichment”). There were 166 significant GO ontologies associated with genes induced during acclimation. Ontologies associated with stress-related processes, and nitrogen use and protein production were the two most over-represented groups (Figure 7).

Genes and regulatory processes repressed during cold acclimation processes have been less studied. There were 1906 arabidopsis gene models that were down-regulated during acclimation (Appendix A, page “gene_list”). Gene set enrichment analysis identified 132 significantly over-represented promoter motifs—the vast majority were ERF/AP2 transcription factor binding sites. Interestingly, CBF1, 2, and 4 binding sites were specifically listed as were DREBs 1A, 2, 2C, 19, and 26 (Figure 8 and Appendix A, page “Motif_enrichment”), suggesting that other factors might override the CBF regulon in controlling the expression of these genes or that the CBF family of transcription factors also down-regulates genes during cold acclimation processes. This supported the over-representation of a large number of ERF/AP2 transcription factor-regulated genes detected in ChIP studies. Nine KEGG ontologies were over-represented, and most of them were involved in carbon and fatty acid metabolism processes. This also compliments the observation from the clusters of significantly over-represented GO ontologies among genes that were either up- or down-regulated during acclimation in both varieties (Figure 7). Stress-related, nitrogen use/protein production-related, transport/signaling-related, and circadian/light quality-related ontologies were generally more prevalent among genes that were up-regulated during acclimation in both varieties than among genes that were down-regulated. Conversely, ontologies related to photosynthetic processes and carbon and lipid metabolism as well as ontologies associated with growth and development were more prevalent among genes that were down-regulated following acclimation. 

### 2.4. Differential Regulation of Circadian Responses, Photosynthetic Processes, and Hormones Associated with Abiotic Stress Are Associated with Deacclimation

There were only 371 and 162 arabidopsis gene models (Appendix A, pages labeled “gene_list”, and Appendix A, yellow highlighted cells in column “AGI” on page “4A KS4666 vs. 1D KS4666”) that were up- or down-regulated, respectively, in both varieties following a shift from cold-acclimated to one-week deacclimated conditions. Gene set enrichment analysis indicated that most of the observed processes involved in photosynthesis and carbon metabolism prevalent among down-regulated genes as plants became acclimated reversed and were turned back on during the deacclimation process (Figure 7 and Appendix A, pages labeled “GO_enrichment”). Likewise, many of the gene ontologies associated with stress and circadian/light quality responses that were more prevalent among genes up-regulated during acclimation were more prevalent among genes that were down-regulated during deacclimation. Genes involved in carbon metabolism and photosynthesis, which were generally down-regulated during acclimation, were similarly represented among up- and down-regulated genes during deacclimation. Intriguingly, genes associated with nitrogen, protein and RNA production, transport and signaling, and growth and development-related ontologies showed a similar trend regarding the proportion of genes that were up- or down-regulated during both acclimation and deacclimation processes, potentially suggesting that these processes are not reversed during deacclimation.

### 2.5. Differences between Deacclimated and Non-Deacclimated Varieties Highlight Defense Responses in the Deacclimation-Sensitive Variety KS09068B-5-1

Gene set enrichment analysis was also performed on genes that differentiated between the deacclimated state among varieties KS09068B-5-1 and KS4666 (excluding genes that were differential between these two varieties under either unacclimated or cold-acclimated conditions—Appendix A yellow highlighted cells in column “AGI” on page 1D KS09068B-5-1 vs. 1D KS4666“ and in Appendix A, pages labeled as “gene_list”) as these likely represent genes that differentiate specifically between the deacclimation rate in these two varieties. Among the genes that were up-regulated in variety KS4666 relative to KS09068B-5-1 during deacclimation, ontologies associated with stress and hormones growth and development were more prevalent (Figure 9). Among the genes more highly expressed in variety KS09068B-5-1, ontologies associated with nitrogen metabolism/protein production, signaling, and transport were more prevalent. However, the stress-related gene ontologies involved in flavonoid and phenylpropanoid biosynthesis and response were more prevalent among genes more highly expressed in KS4666 during deacclimation, suggesting a possible mechanism for KS4666 maintaining higher levels of freezing tolerance (Appendix A).

### 2.6. GWAS Identified Multiple Loci Associated with Freezing Tolerance after Deacclimation

GWAS was run with multiple models to identify loci associated with freezing tolerance following deacclimation at 20 °C days and 10 °C nights for three days (Appendix A). We identified four markers (Figure 10) associated with freezing tolerance following deacclimation using one of the best models from our initial analysis (mixed linear models with a PCA of 17 and kinship). Two additional markers of interest were implicated through follow-on analyses. Local linkage decay helped identify genes leaked to these markers (Table 1).

The most significantly associated marker (S1_750314091) maps to position 37,666,286 on chromosome Ann_random from the *B. napus* reference assembly [15]. This region of the genome has a fast linkage decay rate with loss of linkage occurring within 100 bases in this population [16]. There are no known *B. napus* gene models within this narrow region, but the marker does fall in the middle of a characterized EST (DMBras.unigeneT00180796001).

Another associated marker (S1_698489879) maps to chromosome C06_random at position 1114643. Linkage decay occurs within 20 Kb on either side of this marker, and the marker falls into the *B. napus* gene model BnaC06g41810D. This is a non-conserved gene of unknown function. However, there are three additional genes within the associated interval. One is a gene of unknown function (BnaC06g41820D). Another (BnaC06g41830D) has homology to AT1G52720 of unknown function with a gene product that is located in the chloroplast and induced by methyl jasmonate in arabidopsis. The final gene (BnaC06g41840D) in this interval has similarity to AT1G52730, which encodes a Transducin/WD40 repeat-like superfamily protein that may be responsive to TOR kinase signaling [17].

Marker S1_770664926 is also associated with the deacclimation response and is located on chromosome Cnn_random at position 9358695. Linkage is lost within 30kb in this region. Fifteen gene models are present in this interval. These include BnaCnng10210D, BnaCnng10220D, BnaCnng10230D, BnaCnng10240D, BnaCnng10250D, BnaCnng10260D, BnaCnng10270D, BnaCnng10280D, BnaCnng10290D, BnaCnng10300D, BnaCnng10310D, BnaCnng10320D, BnaCnng10330D, BnaCnng10340D, and BnaCnng10350D. These are similar, respectively, to a DHHC-type zinc finger family protein (AT5G04270), a thioredoxin gene WCRKC2 (AT5G04260), a cysteine protease (AT5G04250), ELF6 (AT5G04240), SYTC (AT5G04220), AtMC9 (AT5G04200), AtMC9 (AT5G04200), PKS4 (AT5G04190), ATACA3 (AT5G04180), Calcium-binding EF-hand family protein (AT5G04170), Nucleic acid-binding, OB-fold-like protein (AT1G14800), Nucleotide-sugar transporter family protein (AT5G04160), Galactose oxidase/kelch repeat superfamily protein (AT5G49000), GLU1 (AT5G04140), and a Phosphoglycerate mutase family protein (AT5G04120). The marker falls into the PSK4 gene. Further, the probable Calcium-binding EF-hand family protein encoded by BnaCnng10250D was significantly up-regulated during deacclimation in variety KS09068B-5-1.

There are 20 gene models present (from BnaC03g68090D to BnaC03g68280D) in the region between chrC03 positions 57,959,505 and 58,130,484 that make up the area within the limits of linkage decay surrounding marker S1_38158858. This marker falls within a homologue of MTO2 (AT4G29840), a gene that is cold-regulated in arabidopsis. Next to this gene is a homologue of VERNALIZATION INDEPENDENCE 3 (BnaC03g68260D), which could also be involved in temperature responses. An analysis of all the models indicates that a marker on A06 could also be of interest (Appendix A).

The marker S1_122523793 falls into a region with high linkage decay but is located in the 17th exon of a large calcium-dependent lipid-binding family gene (BnaA06g04060D) that is similar to the arabidopsis gene AT1G48090. Several other closely linked markers located in the same gene are also highly associated with deacclimation.

Finally, the marker that was found to be significant in three additional programs with various models was S1_757283058. This marker falls in a region with very rapid linkage decay (less than 200 bases before linkage is lost through most of the region) but is located in the third exon of gene model BnaAnng39250D. This gene is homologous to AT3G11510 encoding for a stress-inducible ribosomal protein S11 family protein that is down-regulated following various osmotic stresses including cold stress in arabidopsis.

## 3. Discussion

### 3.1. Genetic Analysis of Deacclimation Processes Identifies Numerous Candidate Genes

In this study, we identified a treatment that differentiates divergent varieties of winter *B. napus* for deacclimation responses. Interestingly, a comparison between the freezing damage between of the fully acclimated varieties and the freezing damage of the corresponding varieties following deacclimation showed minimal correlation (r^2^ = 0.21) [18]. This suggests that there are fundamental differences controlling the freezing tolerance following these two treatments (acclimated and deacclimated). Of the 397 lines in our winter *B. napus* diversity panel that were cold-acclimated, a full range of phenotypes was observed including varieties that were killed by freezing stress after just three days of deacclimation, to varieties that appeared to maintain their cold-acclimated freezing tolerance following deacclimation. This range of phenotypes made it possible to perform a GWAS on the data and identify possible loci associated with altered deacclimation responses. To the best of our knowledge, this represents the first genetic investigation of deacclimation in plants. Surprisingly, there has been very few studies on the mapping of loci controlling cold acclimation or freezing tolerance in *B. napus* or arabidopsis. Further, analysis of loci identified in other studies as related to freezing tolerance in *B. napus* did not identify any of those in this study. This is likely due to differences in how freezing tolerance was phenotyped and, as noted above, differences in the physiology controlling freezing tolerance in deacclimated relative to fully acclimated states. Likewise, none of the candidate genes highlighted in this study were noted in GWAS or QTL mapping studies involving acclimation processes in arabidopsis [14,19]. Since freezing survival following deacclimation could be influenced by basal (unacclimated) freezing tolerance of any given line, we cannot rule out that markers associated with freezing tolerance following deacclimation might also be associated with basal freezing tolerance. Further studies are needed to determine if any of our markers play a role in such basal freezing tolerance in *B. napus*.

Establishing parameters for the significance of associations between the phenotype and genotype is somewhat arbitrary. In this study, the top four were chosen (*p* < 10^−4^, which is the significant value for the top 0.01% of the markers). The most associated marker (S1_750314091) fell into an area of the chromosome with very rapid linkage disequilibrium decay and thus the number of possible candidate genes was limited. This marker did not fall into a recognized gene model, but did fall into a transcribed but non-coding sequence and thus could still have some functional significance. Although some non-coding RNAs are important regulatory elements and are often conserved, this transcribed region does not appear to be conserved in arabidopsis. It will be interesting to see if this locus is near any quantitative trait loci (QTLs) for deacclimation response in crosses between varieties that both differ in their deacclimation response and have variability in this marker. Without further indications for an association with deacclimation responses, it seems likely that this marker may represent a false positive association.

Other potential candidate genes were identified near the second most significant marker. Although this marker falls into a recognized gene model, our transcriptome analysis indicated this gene model was not transcribed, which may explain the lack of homology to any known arabidopsis genes. However, two genes found within the interval of linkage decay are represented by arabidopsis genes and could potentially be involved in deacclimation processes. One (AT1G52720) encodes a gene of unknown function that appears to be regulated by the stress-responsive hormone methyl jasmonate [20] and is strongly down-regulated in the roots of arabidopsis following exposure to cold. It is also up-regulated in *B. napus* following deacclimation (Appendix A). The other is similar to a transducing/WD40 repeat containing gene AT1G52730, which can be precipitated as part of the TOR Kinase complex in arabidopsis [17]. SNF1-related kinases along with TOR kinase are involved in sensing the nutrient and stress status of plants and integrate those signals to balance the decision to grow or commit resources to defense [21]. Interestingly, several SFN1-related kinase genes were significantly down-regulated following deacclimation in two of the winter *B. napus* varieties examined here. Additionally, growth inhibition is often associated with induction of freezing tolerance, but it is not yet known if growth cessation is needed for cold acclimation to occur. Thus, it may be probable that TOR-regulated growth inhibition provides a mechanism for controlling variation in freezing survival following deacclimation. It will be interesting to determine if the more rapidly or completely deacclimating variety KS09068B-5-1 has more vigorous growth than KS4666 at 10 °C. However, either of these genes could be potential candidates for impacting deacclimation rates/responses in winter *B. napus* and are worthy of further study.

Additional gene models were also identified within the linkage decay interval for the significantly associated marker on chromosome Cnn_random. Thirteen gene models were present in this region with several having potential roles in deacclimation processes. The significant marker mapped to the gene encoding a likely orthologue of PHYTOCHROME KINASE SUBSTRATE 4 (PSK4). This gene is involved in phytochrome signaling and is up-regulated by both auxin and brassinolide, suggesting a possible role in growth regulation. Phytochrome signaling is integral in both gating cold-regulated gene expression [22] and flower development. Additionally, it has been proposed that the vernalization response reduces the ability of plants to maintain freezing tolerance [23]. Another gene of possible interest identified within the linkage decay interval is EARLY FLOWERING 6 (ELF6). Again, this gene has a role in regulating the circadian cycle, which impacts acclimation processes [24]. Additionally, gene set enrichment analysis identified the binding site for RELATED TO EARLY FLOWERING 6 (REF6) as the most significant over-represented sequence in the promoters of deacclimation-responsive genes in this study. Finally, the gene model encoding a calcium-binding EF-hand family protein related to AT5G04170 in this interval was differentially expressed only in the deacclimation-sensitive variety KS09068B-5-1. Given that calcium channels have been associated with low-temperature perception [5], this gene is of particular interest. There are other genes as well in this region that could be responsible for the association of this locus with deacclimation responses in winter *B. napus*. Arabidopsis lines with mutations in two of the genes (PSK4 and the orthologue of AT5G04170) identified in this linkage decay interval are currently being tested for their ability to withstand freezing following deacclimation.

Twenty different gene models were found within the linkage decay region for a marker on chromosome C03. However, the marker itself does fall in the coding region of a probable orthologue of METHIONINE OVER-ACCUMULATOR 2 (MTO2). This protein encoded by this gene is located in the chloroplast membrane, and mutants of this gene accumulate as much as 20× the level of methionine as wild type plants. Increases in neutrally charged amino acids such as proline have long been associated with increased freezing tolerance [2], and methionine transport in *B. napus* increased during cold acclimation induced by ABA [25].

Finally, the marker on chromosome Ann_random identified by the other three models was among the top 500 markers identified by the best model in TASSEL5 (with a *p* value of 1.8 × 10^−3^). This marker fell into the third exon of a gene encoding a ribosomal protein S11 family protein. There are three other similar genes in arabidopsis, but this gene appears to be orthologous to AT3G11510, based on the synteny of surrounding genes in this region of the Ann_random chromosome. Although this gene did not appear to be differentially regulated in our RNAseq analysis, it was down-regulated by several osmotic-related abiotic stresses in arabidopsis. It will be interesting to see if mutations in this gene will impact deacclimation processes in arabidopsis. These studies have been initiated.

### 3.2. RNAseq Identifies Genes and Processes Previously Associated with Freezing Tolerance and Cold Acclimation

Numerous genes were up-regulated by cold acclimation in two winter *B. napus* varieties divergent for deacclimation. Many of these genes are indicative of abiotic stress responses, such as flavonoid and coumarin biosynthetic processes, and regulation of multiple stress response systems and hormones. Likewise, an assessment of down-regulated genes in response to acclimation includes numerous processes involved in photosynthesis and growth. All are consistent with previous studies on cold acclimation in plants [7,26,27,28]. Surprising, however, was the observed lack in induction of common COR genes usually observed during cold acclimation. Several, such as the COR15a and COR15b, were down-regulated during acclimation in our dataset and induction of CBF orthologues was also not observed. The reason for this observation is unclear but the CBF regulon can be induced by factors other than cold stress. For example, touch can rapidly induce the CBF regulon [29], and it is possible that CBF may have been induced under greenhouse conditions which were occasionally quite windy due to cooling fans employed to maintain temperatures. Although CBF is induced rapidly by the cold, it also shows a gradual drop in expression over extended periods of cold [30], thus it is also possible that after four weeks, the level of CBF dropped below baseline levels. Interestingly, CBF binding sites (CCGAC) were over-represented in the promoters of genes that were up-regulated by acclimation, suggesting some genes normally up-regulated by the CBF regulon remained up-regulated—even though the CBF regulon was not.

Among the genes and processes that are down-regulated during cold acclimation, photosynthesis and growth processes are most prominent. Even though the acclimated plants were provided with light, photosynthesis in the cold can damage plants due to imbalances between photosystems that can result in the production of reactive oxygen species [31]. Thus, it is not surprising that photosynthetic processes were down-regulated in both varieties following acclimation. Likewise, growth has been negatively associated with a plant’s ability to cold-acclimate [5,32], and it is not surprising that acclimated plants would down-regulate growth processes. With the exception that activation of the CBF regulon was lacking, the observed changes in gene expression following cold acclimation were consistent with previous studies.

### 3.3. Similarities and Differences in Response to Deacclimation between Varieties Highlight Defense Responses and Photosynthetic Processes

In the winter *B. napus* varieties used in this study, deacclimation generally had the opposite response to acclimation. Thus, the observed reactivation of photosynthesis during deacclimation was not dissimilar to what others have observed [33,34]. However, we did not see continued up-regulation of photosynthetic responses as reported [33], during acclimation. Indeed, we observed substantial down-regulation of such genes that were then generally up-regulated again following deacclimation (Figure 8). These differences appeared to be more enhanced in the deacclimation-sensitive variety KS09068B-5-1 than in KS4666. One might expect that variety KS09068B-5-1 would more closely resemble the unacclimated state, since it has lost most of its freezing tolerance within one week at 10 °C. Thus, we might expect KS09068B-5-1 to show higher expression of photosynthesis genes than KS4666. The genes that were more highly expressed in KS09068B-5-1 during deacclimation did show an over-representation of photosynthetic genes and decreased expression in flavonoid and phenylpropanoid biosynthetic processes that might be associated with enhanced ability to deal with oxidative stress. Interestingly, a strong over-representation among genes involved in defense, specifically biotic stress responses indicative of SA signaling, was observed among genes that were more highly expressed in deacclimated KS09068B-5-1 relative to KS4666. This observation may suggest that the fully deacclimated variety was less able to compensate for the increase in oxidative stress that photosynthesis might generate at chilling (10 °C) temperatures.

Another surprising observation was that many AP2/ERF binding sites including that of CBF were highly over-represented in genes that were less highly expressed in KS09068B-5-1 compared to KS4666. Again, this observation may reflect the inability of KS09068B-5-1 to maintain or increase the level of freezing tolerance under extended periods at low but not fully acclimating temperatures. It should be noted that we have not assessed the timing of vernalization processes attributed to either of our tested varieties. Although neither variety showed obvious differences in growth or flowering during the deacclimation process, it is possible that some of the differences in gene expression and freezing tolerance could be due, in part, to potential differences in the developmental state of either variety. Further study will be needed to determine the role—if any—of flowering capacity following four-week acclimation processes.

Due to the gene set enrichment program we used, the analyses were performed on the arabidopsis orthologues of the *B. napus* genes. Although not ideal, virtually all of the *B. napus* genes that were annotated had similar annotations to their arabidopsis counterparts and thus would have the same gene ontology terms associated with them. However, an advantage of using the arabidopsis gene orthologues for enrichment analysis is that it provides access to much larger databases of confirmed transcription binding sites and ChIPseq analyses. Although some binding sites might not be conserved between *B. napus* and arabidopsis, there are ample examples in the literature where the binding sites are conserved between arabidopsis and species considerably more distantly related than *B. napus*. Thus, we are reasonably confident in the data we present, although further studies are needed to test hypotheses developed from our results.

## 4. Materials and Methods

### 4.1. Deacclimation Studies and Freezing Stress Assessment

Plants were grown in 800 mL pots and potting soil under greenhouse conditions (~22 °C with a 16-h photoperiod and supplemental halogen lighting as needed) to the 6–8 leaf stage (approximately 3–4 weeks after planting). Plants were then transferred to a 5 °C cold acclimation chamber with a 12-h photoperiod (with supplemental full spectrum LED lighting—Lumigrow LU50001) for 4 weeks. Cold-acclimated plants were then moved to a growth chamber for deacclimation at a constant temperature of 5, 10, or 15 °C for various times as indicated for assessment of deacclimation response and to collect tissue samples as indicated below for RNAseq analysis. For GWAS studies, acclimated plants were deacclimated by altering the acclimation chamber temperature to a 20 °C light 12-h and 10 °C dark 12-h regime. Plants were then subjected to a freezing stress to quantify the loss of freezing tolerance during the deacclimation treatments. For assessing loss of acclimation in the two varieties chosen for RNAseq analysis, freezing stress was initiated at 12:00 PM (noon) at 15 °C, then ramped down to 5 °C over five hours, then ramped down to the freezing temperature (−10 °C) over eleven hours (4:00 AM), then held at that temperature for 4 h before ramping back up to 0 °C over 1 h, and then back up to 15 °C by 12:00 PM the following day (lights were turned on over the last hour of freezing temperatures). Plants were moved to a greenhouse and were scored for visual damage on a 0–3 scale with 0 being death, 1 having >50% foliar damage but maintaining at least one living meristem, 2 having between 50% and 10% foliar damage, and 3 having 0–10% foliar damage after a one-week recovery period. Three to nine plants for each variety were treated in each experiment and were randomized prior to freezing to reduce the possibility of variation in pot position within the freezing chamber. Experiments were replicated at least twice. For the GWAS, freezing was initiated with a ramp down in temperature from 20 to −10 °C over a five-hour period staring at 12:00 PM in the dark and held at −10 °C for four hours. Plants were then ramped back up to 20 °C by 12:00 the next day and then moved to the greenhouse for recovery and visual damage scores which were taken two weeks later.

### 4.2. RNAseq Analysis

Winter *B. napus* varieties KS4666 and KS09068B-5-1 showed significant differences in deacclimation intensity at 10 °C and were chosen for RNAseq analysis. Deacclimation was complete after seven days at 10 °C for variety KS09068B-5-1. However, variety KS4666 maintained significant freezing tolerance even after two weeks at 10 °C. Thus, these two varieties were chosen for RNAseq analysis (NCBI accession # PRJNA560411). The distal 5 cm of the largest green leaves from three plants per treatment were harvested and pooled for each sample. Three samples were collected from each treatment (unacclimated, acclimated for four weeks, deacclimated for one week, deacclimated for two weeks) for both KS09068B-5-1 and KS4666.

RNA was extracted from the leaf samples and directional PolyA+ cDNA libraries were prepared from each using the NEBNext^®^ Ultra™ Directional RNA Library Prep Kit for Illumina^®^ as per the manufacturer’s protocols (New England BioLabs, Inc., Ipswich, MA, USA). Libraries were size-selected for fragments between 200 and 400 bases using a Pipin-prep system and checked for quality and quantity using bioanalyzer and qubit systems, respectively. Libraries were pooled and sequenced by BGI America. The resulting reads were uploaded into the Cyverse Discovery Environment [35]. Sequences were trimmed for quality (quality score values > 20) and contaminating primers were removed using the program HTProcess_trimmomatic_0.33. Sequences were mapped to the reference genome (Brassica_napus_v4.1.chromosomes.fa with annotation file Brassica_napus.annotation_v5.gff3.gz) [15], quantified, and differential expression between each pair of treatments/cultivars was performed using the tuxedo suite of programs in the CyVerse Discovery Environment. Annotation of the *B. napus* genes was derived from the file. BlastX was used to identify the most likely arabidopsis homologue for each of the *B. napus* genes. Differentially expressed genes were subjected to gene set enrichment analysis (GSEA) based on the functions of the closest arabidopsis homologues using the suite of analyses available from the Plant Regulomics website [36].

### 4.3. Genome-Wide Association Study

A well-characterized diversity panel of primarily winter *B. napus* varieties [16] composed of 397 varieties was used in this studied. Briefly, this population is diverse with a minimal population structure with a K vs. Ln likelihood plot showing a drop between K-values of 2 and 4 and a neighbor-joining tree indicated as few as 6 major branches. Following imputation, 251,575 high-quality single-nucleotide polymorphism (SNP) markers with an average distance between them of 3377 bases were available for associations. After acclimation, deacclimation, and freezing, plants were scored for damage using the visual damage scale described above. This experiment was run with three technical replicates in a random complete block design and the damage scores for each variety were averaged. Each experiment was repeated three times for a total of nine data points for each variety. The program Tassel5 was used with multiple models including general linear model (GLM) under naïve conditions with 1000 iterations, GLM with a principal component (PCA) of 1, GLM with a PCA of 17 (explaining 25% of the variability), mixed linear models (MLMs)—all with kinship, MLM with a PCA of 0, 1, and 17, and all of the above with or without a covariate of the population structure for a K value of 3. GLM with a PCA of 17 and no population covariate and an MLM with a PCA of 17 with kinship and no population covariate appeared to be the best models based on the MSD method [37]. Additionally, the programs Fixed and random model Circulating Probability Unification (FarmCPU) [38] and BLINK [39] under naïve conditions, and SUPER [40] with a PCA of 3 were also run. Candidate genes surrounding 4 of the top-most associated markers from the Tassel5 analyses and one marker that was indicated as significant by all three of the additional programs were chosen. The region of interest surrounding each marker was based on the physical distance where the local linkage decay (R2) crossed a threshold of 0.2 (see [16] for details).

## 5. Conclusions

This work represents the first genetic analysis of deacclimating processes in *B. napus*. We used a GWAS approach and identified several genes that could be candidates for altering the rate or intensity of the deacclimation process. Many of the candidate genes have roles in light signaling, circadian responses, and regulation of growth. This observation is consistent with some of these processes being over-represented among genes that were differentially expressed during cold acclimation and deacclimation in *B. napus*. Further studies are underway to assess the impact of these candidate genes in the deacclimation process.

## Figures and Tables

**Figure 1 ijms-21-09148-f001:**
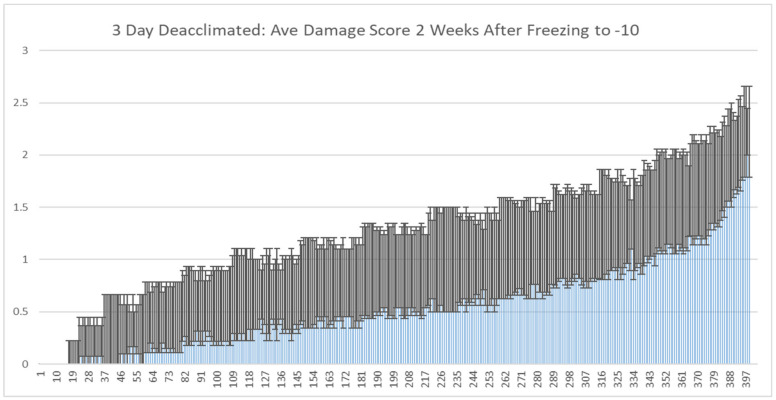
Average damage scores of 397 primarily winter *B. napus* varieties in our diversity panel following four weeks of cold acclimation (5 °C) then three days of deacclimation (22 °C day 10 °C nights) followed by freezing to −10 °C for four hours and then recovery for two weeks before visual damage scores were taken. Visual damage scores (y-axis) were rated as follows: 3 = 0–10% leaf damage, 2 = 10–50% damage, 1 = 50–99% damage but still showing some regrowth potential after 2 weeks, 0 = dead with no regrowth after 2 weeks. *n* = 9, error bars represent the SE (standard error).

**Figure 2 ijms-21-09148-f002:**
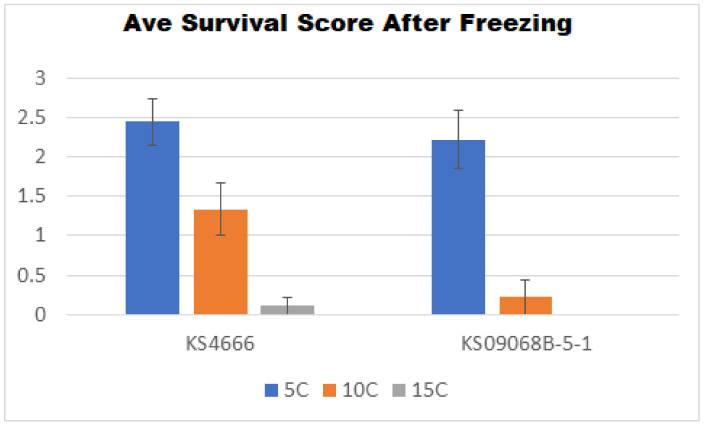
Average freezing damage scores (y-axis) for two divergent *B. napus* varieties (KS4666 and KS09068B-5-1) in fully acclimated plants at 5 °C, or following deacclimation at 10 °C, or 15 °C for three days after a 4-week acclimation period (5 °C). Visual damage scores are rated as follows: 3 = 0–10% leaf damage, 2 = 10–50% damage, 1 = 50–99% damage but still showing some regrowth potential after 2 weeks, 0 = dead with no regrowth after 2 weeks. *n* = 9, error bars represent the SE.

**Figure 3 ijms-21-09148-f003:**
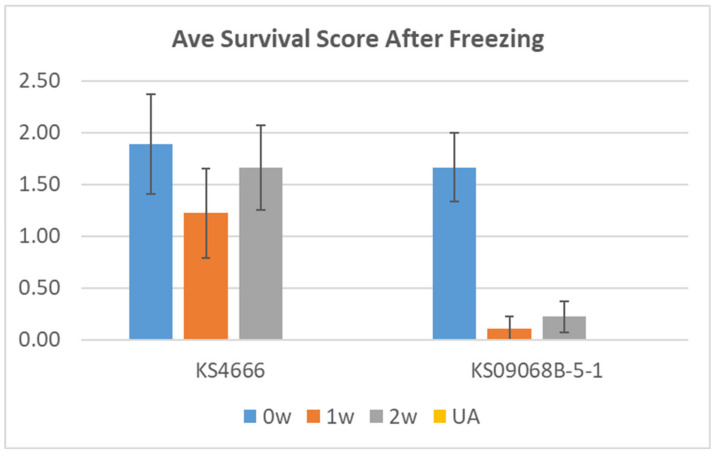
Average freezing damage scores (y-axis) for two divergent *B. napus* varieties (KS4666 and KS09068B-5-1) following deacclimation at 10 °C for 0, 1, or 2 weeks with or without (UA) a 4-week acclimation period (5 °C). Visual damage scores are rated as follows: 3 = 0–10% leaf damage, 2 = 10–50% damage, 1 = 50–99% damage but still showing some regrowth potential after 2 weeks, 0 = dead with no regrowth after 2 weeks. *n* = 9, error bars represent the SE.

**Figure 4 ijms-21-09148-f004:**
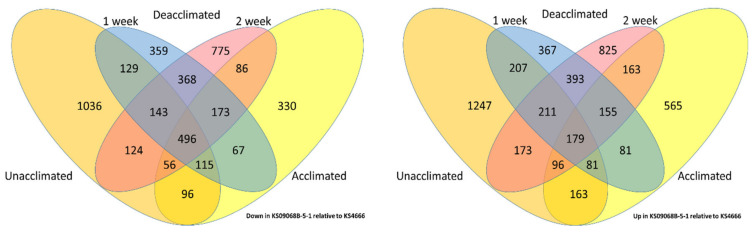
Venn diagram showing the number of genes that were differential between the two varieties following the different treatments.

**Figure 5 ijms-21-09148-f005:**
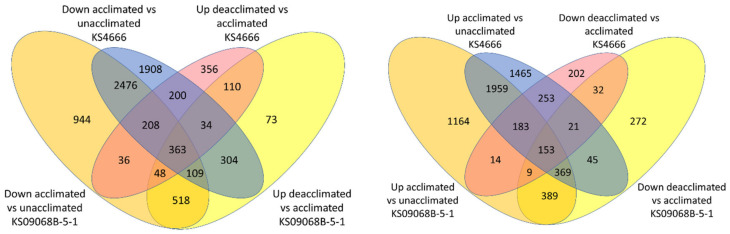
Genes up-regulated or down-regulated in acclimated relative to deacclimated (one week) in both varieties.

**Figure 6 ijms-21-09148-f006:**
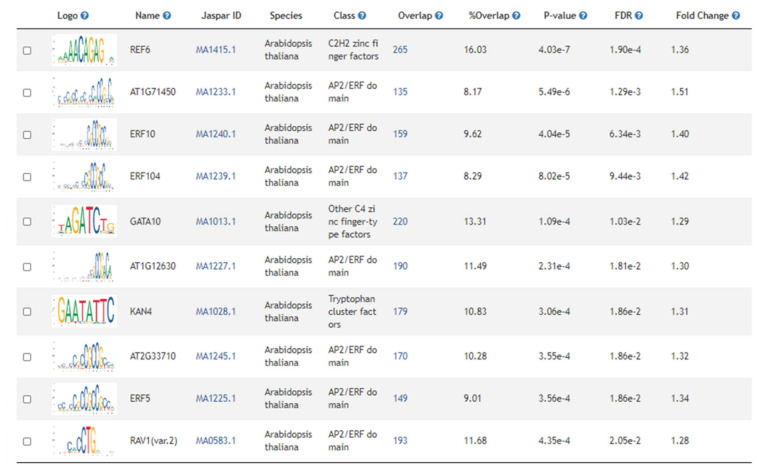
Motif enrichment in genes up-regulated during acclimation in both KS4666 and KS09068B-5-1. Motifs are sorted by *p* value, with the most significant at the top.

**Figure 7 ijms-21-09148-f007:**
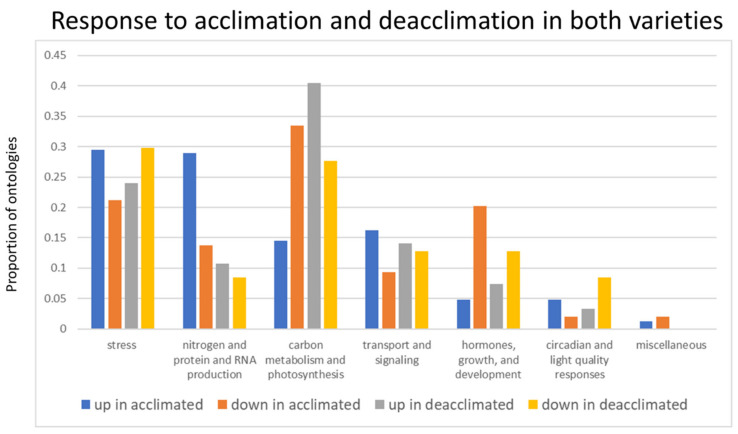
The proportion of ontologies over-represented among the significantly (FDR > 0.05) up- or down-regulated genes in designated clusters during acclimation and deacclimation. All gene ontologies were clustered into seven categories with relevance to cold acclimation processes, and the proportion of ontologies assigned to each cluster out of the total number of gene ontologies identified as significant was determined.

**Figure 8 ijms-21-09148-f008:**
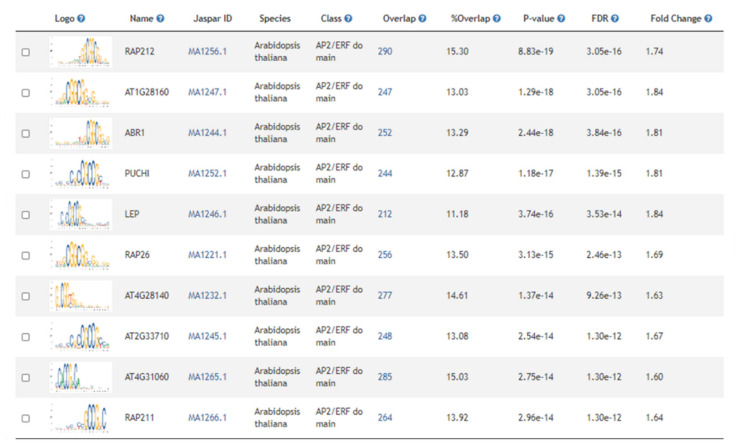
Motif enrichment in genes down-regulated during acclimation in both KS4666 and KS09068B-5-1. Motifs are sorted by *p* value, with the most significant at the top.

**Figure 9 ijms-21-09148-f009:**
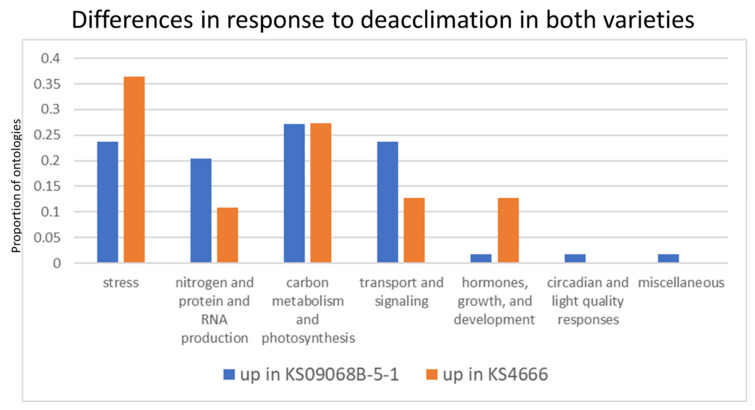
The proportion of ontologies over-represented among genes up-regulated in KS09068B-5-1 or up-regulated in KS4666 relative to the other variety during deacclimation. All gene ontologies were clustered into seven categories with relevance to cold acclimation processes, and the proportion of ontologies assigned to each cluster out of the total number of gene ontologies identified as significant was determined.

**Figure 10 ijms-21-09148-f010:**
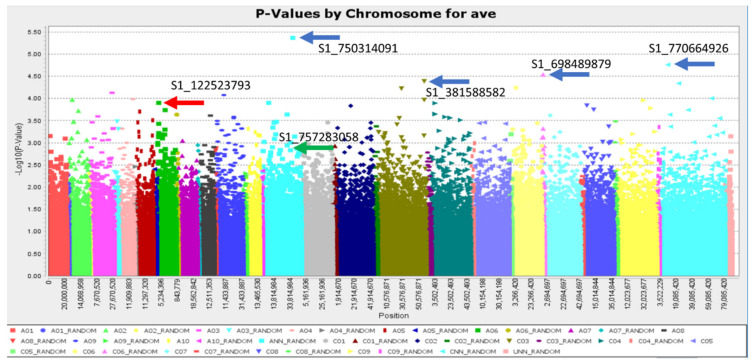
Manhattan plot from the GWAS analysis using a mixed linear model with kinship and a principle component of 17 showing the top 4 markers (blue arrows), the marker on A06 that had low average *p* values across all models (red arrow), and a marker that was consistently identified as a top marker on additional tests (BLINK, FarmCPU, and SUPER) (green arrow).

**Table 1 ijms-21-09148-t001:** List of significant markers, the local linkage decay range, all gene models located within that range, their chromosomal position, the associated arabidopsis orthologue, and the putative gene function for each gene model. The bold *B. napus* gene model indicates the gene that was differentially expressed in response to deacclimation treatment. Highlighted map positions indicate the genes of which the significant marker maps.

Marker	Position	LD Interval	*B. napus* Gene Model	Gene Model Map Position	Arabidopsis Orthologue	Probable Gene Function
S1_750314091	Ann_Random 37666286	37666186-37666386	DMBras.unigeneT00180796001	**109261544 to 109263434**	NA	NA
S1_698489879	C06_random 1114643	1094643-1134643	BnaC06g41810D	**1113495 to 1114219**	NA	NA
			BnaC06g41820D	1126313 to 1126528	NA	NA
			BnaC06g41830D	1133565 to 1134202	AT1G52720	unknown function
			BnaC06g41840D	1135605 to 1138056	AT1G52730	Transducin/WD40 repeat-like superfamily protein
S1_770664926	Cnn_random 9358695	9328695-9388695	BnaCnng10210D	9333223 to 9335001	AT5G04270	DHHC-type zinc finger family protein
			BnaCnng10220D	9335248 to 9336428	AT5G04260	WCRKC thioredoxin 2
			BnaCnng10230D	9336609 to 9338164	AT5G04250	Cysteine proteinases superfamily protein
			BnaCnng10240D	9339664 to 9344492	AT5G04240	transcription factor jumonji (jmj) family protein
			**BnaCnng10250D**	9353589 to 9356426	AT5G04220	deacclimation-regulated Calcium-dependent lipid-binding (CaLB domain)
			BnaCnng10260D	9356627 to 9356920	AT5G04200	metacaspase 9
			BnaCnng10270D	9356994 to 9357596	AT5G04200	metacaspase 9
			BnaCnng10280D	**9358250 to 9359464**	AT5G04190	phytochrome kinase substrate 4
			BnaCnng10290D	9360809 to 9362430	AT5G04180	alpha carbonic anhydrase 3
			BnaCnng10300D	9365237 to 9367298	AT5G04170	Calcium-binding EF-hand family protein
			BnaCnng10310D	9368412 to 9369733	AT1G14800	Nucleic acid-binding, OB-fold-like protein
			BnaCnng10320D	9372342 to 9374120	AT5G04160	Nucleotide-sugar transporter family protein
			BnaCnng10330D	9374957 to 9376012	AT4G36840	Galactose oxidase/kelch repeat superfamily protein
			BnaCnng10340D	9376962 to 9384925	AT5G04140	glutamate synthase 1
			BnaCnng10350D	9386245 to 9387697	AT5G04120	Phosphoglycerate mutase family protein
S1_38158858	C03 57859505	57759505-5799505	BnaC03g68090D	57742606 to 57744811	AT2G19090	Protein of unknown function (DUF630 and DUF632)
			BnaC03g68100D	57751536 to 57752049	AT4G30074	low-molecular-weight cysteine-rich 19
			BnaC03g68110D	57761207 to 57764215	AT4G30060	Core-2/I-branching beta-1,6-N-acetylglucosaminyltransferase protein
			BnaC03g68120D	57765406 to 57766578	AT4G30010	LOCATED IN: mitochondrion, plastid
			BnaC03g68130D	57768296 to 57772432	AT4G29960	LOCATED IN: plasma membrane
			BnaC03g68140D	57772494 to 57774352	AT4G29950	Ypt/Rab-GAP domain of gyp1p superfamily protein
			BnaC03g68150D	57774688 to 57778115	AT4G29940	pathogenesis related homeodomain protein A
			BnaC03g68160D	57788448 to 57789017	AT4G29930	basic helix-loop-helix (bHLH) DNA-binding superfamily protein
			BnaC03g68170D	57789892 to 57792284	AT4G29930	basic helix-loop-helix (bHLH) DNA-binding superfamily protein
			BnaC03g68180D	57793363 to 57793613	NA	NA
			BnaC03g68190D	57793691 to 57793750	NA	NA
			BnaC03g68200D	57816672 to 57819918	AT4G29920	Double Clp-N motif-containing P-loop nucleoside triphosphate hydrolases
			BnaC03g68210D	57826932 to 57827135	AT4G29905	Unknown conserved
			BnaC03g68220D	57833058 to 57834309	AT5G63820	Protein of unknown function (DUF626)
			BnaC03g68230D	57839948 to 57841025	AT5G63820	Protein of unknown function (DUF626)
			BnaC03g68240D	57841738 to 57843321	AT5G63820	Protein of unknown function (DUF626)
			BnaC03g68250D	**57858635 to 57860206**	AT4G29840	Pyridoxal-5′-phosphate-dependent enzyme family protein
			BnaC03g68260D	57860353 to 57861992	AT4G29830	Transducin/WD40 repeat-like superfamily protein VIP3
			BnaC03g68270D	57956109 to 57956560	AT4G29700	Alkaline-phosphatase-like family protein
			BnaC03g68280D	57961823 to 57962718	AT4G29690	Alkaline-phosphatase-like family protein
S1_122523793	A06 2474952	2473952-2475952	BnaA06g04060D	**2451590 to 2477036**	AT1G48090	calcium-dependent lipid-binding family protein
S1_757283058	Ann_random 44635253	44635153-44635353	BnaAnng39250D	**44634379 to 44636475**	AT3G11510	Ribosomal protein S11 family protein

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
