# Peer review of "Genome-Wide Association Studies and Transcriptome Changes during Acclimation and Deacclimation in Divergent Brassica napus Varieties"

_ijms, 2020, doi:10.3390/ijms21239148_

Round 1

Reviewer 1 Report

Comments to authors:

Line 1: Most winter B. napus varieties are not of canola quality due to their high erucic acid and glucosinolate contents. Can the authors confirm if they carried out quality analysis on the winter? If not it may be more appropriate to rephrase ‘winter canola varieties’ as ‘Winter B. napus varieties’ throughout the MS.

Line 42-44: Provide reference that reports winter varieties flower faster than spring type Brassica napus.

Line 54-italicise Arabidopsis thaliana

Line 190: The 5820 genes differentially expressed should be clearly identifiable in supplementary file 2. Readers should not have to extract the information by themselves.

Line 192: Similarly, the authors should provide a list of the 1706 Arabidopsis gene models. Readers should not find it difficult to

Line 209: I could not find the list of the 1970 Arabidopsis gene models in Supplementary File 4. Did the authors provided it elsewhere?

Line 210: the supplementary file 4 contains 187 and not 140 significantly over-represented promoter motifs. Can the authors re-check?

Line 242: Which file shows the 404 and 175 gene models? I cannot find them in supplementary files 5 and 6.

Line 270: I did not find supplementary file 8 in the re-submitted MS

Line 279: I could not download supplementary file 9 in the re-submitted MS

Line 278:321: The authors failed to address the main concerns in my original and follow-up reviews. The authors need to provide the results of all the tested models and how were the TAs identified? The Manhattan plot shows that most of the significant MTAs were below LOD=4. Indeed only one marker out of 3377 (original no. 251,575) is above LOD=5.0. The authors need to carry out further analysis to confirm this data. In addition, the gene information should be summarized in a table showing gene name, chromosome, position and species (e.g. B. napus, B. rapa, B. oleracea, Arabidopsis thaliana etc.)

Line 279 and Figure 10: There are six red arrows. The authors stated four MTAs. What the red arrows represent should be stated.

Line 550: check “ a The’ a?

Line 552: Authors should state the source of the 251,575 SNPs and the chromosomal distribution.

Line 553: check ‘baes’-bases?

The authors should change temperature units from C to oC throughout the MS (This was not addressed even after being pointed out in the earlier review).

Horvath et al. is not yet published and so reviewers do not have any information on contents. As such citing it is not relevant.

Author Response

Line 1: Most winter B. napus varieties are not of canola quality due to their high erucic acid and glucosinolate contents. Can the authors confirm if they carried out quality analysis on the winter? If not it may be more appropriate to rephrase ‘winter canola varieties’ as ‘Winter B. napus varieties’ throughout the MS.

That is acceptable as I am not 100% sure all are canola,  I have made that change.

Line 42-44: Provide reference that reports winter varieties flower faster than spring type Brassica napus.

Citation was added.

Line 54-italicise Arabidopsis thaliana

Done

Line 190: The 5820 genes differentially expressed should be clearly identifiable in supplementary file 2. Readers should not have to extract the information by themselves.

These genes were highlighted. The notation of the highlighting is not mentioned in the text of the manuscript as well.

Line 192: Similarly, the authors should provide a list of the 1706 Arabidopsis gene models. Readers should not find it difficult to

These gene lists were refined to include only “expressed genes” but now consistently include genes that were marked as HIDATA that show on/off expression patterns. These gene lists are now included in the supplemental files containing the gene set enrichment analysis and are also highlighted in specific pages in the annotated expression data files. All of the data has been reanalyzed- although these slight changes in the gene lists did not appreciably alter the results or conclusions for most analyses. However there were some differences noted between the deacclimation state of the two varieties- probably because the number of differential genes for that group was rather small after deleting the probable general varietal differences.

Line 209: I could not find the list of the 1970 Arabidopsis gene models in Supplementary File 4. Did the authors provided it elsewhere?

See above

Line 210: the supplementary file 4 contains 187 and not 140 significantly over-represented promoter motifs. Can the authors re-check?

We have rechecked and have now been clear that we are only counting ontologies with an FDR of less than 0.05 as significant.

Line 242: Which file shows the 404 and 175 gene models? I cannot find them in supplementary files 5 and 6.

See above

Line 270: I did not find supplementary file 8 in the re-submitted MS

Hmm. That is odd. The file containing all supplementary files was zipped and tar-balled together for submission. Hopefully they will all be together (again) with this revision.

Line 279: I could not download supplementary file 9 in the re-submitted MS

See comments above.

Line 278:321: The authors failed to address the main concerns in my original and follow-up reviews. The authors need to provide the results of all the tested models and how were the TAs identified? The Manhattan plot shows that most of the significant MTAs were below LOD=4. Indeed, only one marker out of 3377 (original no. 251,575) is above LOD=5.0. The authors need to carry out further analysis to confirm this data. In addition, the gene information should be summarized in a table showing gene name, chromosome, position and species (e.g. B. napusB. rapaB. oleraceaArabidopsis thaliana etc.)

This is a confusing comment since supplemental table 9 has the p-value results of all markers of the models tested, as well as the full data from the two top models. All of the genes within the linkage decay range of each of the 6 markers we considered to be significant are listed in the manuscript results. We have now also made a table showing that data.

Line 279 and Figure 10: There are six red arrows. The authors stated four MTAs. What the red arrows represent should be stated.

I also go on to note one additional marker that looked interesting when all of the models are assessed together, and another that was indicated by showing significance in tree additional analyses. These are now all labeled on figure 8.

Line 550: check “ a The’ a?

Fixed.

Line 552: Authors should state the source of the 251,575 SNPs and the chromosomal distribution.

This is in the submitted manuscript describing the population, the SNP identification process, the imputations checks, and the local chromosomal distribution of the SNPs.

Line 553: check ‘baes’-bases?

Fixed

The authors should change temperature units from C to oC throughout the MS (This was not addressed even after being pointed out in the earlier review).

Fixed.

Horvath et al. is not yet published and so reviewers do not have any information on contents. As such citing it is not relevant.

I expect word on that manuscript within the week.

Reviewer 2 Report

The authors have submitted a revised manuscript in which they have satisfactorily either addressed, corrected or defended comments/concerns raised.

There are no more comments from me.

Author Response

NA

Reviewer 3 Report

Excellent paper.  Recommend publish as is

Author Response

NA

Reviewer 4 Report

In this paper the authors aimed at identifying loci involved in the post-deacclimation tolerance to freezing in a population of Brassica napus accessions, using an association mapping approach. In parallel, on two varieties showing divergent tolerance to freezing, a transcriptome analysis was performed after an appropriate cold treatment to elucidate the molecular mechanisms underlying this trait. A number of markers associated to freezing tolerance were detected, and gene categories differentially expressed between the different treatments and varieties were described and discussed.

The work provides interesting information about the genetic determinants of tolerance traits as well as the physiological mechanisms involved; however, I have some concerns about the experimental workflow, which could be implemented, and in my opinion the paper suffers a few flaws that I suggest to address before publication.

 MAJOR CONCERNS:

  • L512-521: As far as I understood, the freeze treatment to the panel before GWAS was different from that to the two divergent varieties before transcriptome analysis. In my opinion this makes difficult to compare and cross-validate the results. Could you explain why two different treatments were used?
  • L345-350: the authors state that some mechanisms of post-deacclimation tolerance to freezing detected could be related to the basal (pre-acclimation) tolerance. Why not performing a GWAS on the same accessions for pre-acclimation tolerance? The two results could be compared and in this way possible overlapping could be evidenced. In my opinion this would implement the information about the mechanisms detected and would avoid to leave an open question.

GENERAL COMMENTS TO THE MANUSCRIPT

There are some grammar errors which need to be checked and fixed (e.g. L63: “although mutant analyses has implicated” should be “although mutant analyses HAVE implicated”)

Some parts of the manuscript (especially those about the results of the trasncriptome analysis) are quite verbose and difficult to read. English should be revised to make the text more easily readable

Please ensure that all the references are cited in the same way in the text (“et al” is sometimes followed by a dot and sometimes it’s not)

In general, the Celsius degrees should be indicated with ”°C” and not ”C”; the Latin plant names should be written in italics (also in the references)

Below, more specific line by line comments:

L14-16: Affiliations 4 to 6 are not assigned to any of the authors

L86: the reference cited are quite old. With a simple Pubmed search it’s possible to find more recent information about the identification of genes involved in cold tolerance. I suggest to add more up-to-date references

L104: the standard deviations of the phenotypic scores for frost damage are quite high; did you check for possible outliers?

L127 and 135: this is just an aesthetical observation, but I suggest to set fig. 2 and 3 at the same size

L208-220: this part is quite complicate to read; I think it should be simplified (especially L218-220)

L250-253:  genes related to nitrogen, protein and RNA production are much more overexpressed during acclimation than during the following deacclimation, do you have any comments on this?

L261: “stress responses” should be removed

L266: the second comma should be removed

L277: generally speaking about this section, I think that it could be better readable if associated markers were separate in different paragraphs

L280: the reference to figure 8 should probably be to figure 10

L287: “maker” should be “marker”; if I understood correctly “genome” should be “gene”

L317: see L287

L324: in the manhattan plot, the title should be enlarged and fitted to the text; I suggest to include the names of the associated markers aside each arrow

L327: in general, I suggest to write the name of each associated marker after citation

L328-331: if 0.21 is the correlation coefficient (r2), it should be indicated and moved after “correlation”. “(as noted by Chao et al. 2020)” should be at the end of the sentence

L333-336: the verb is missing

L351-353: in the GWAS analysis, did you try a fixed significance threshold (es. 0.05) followed by a correction such as Bonferroni or a correction for multiple tests such as the Benjamini-Hochberg test for False Discovery Rate?

L374: the citation “Leene et al 2019” should be “van Leene et al. 2019”

L380-381: please revise this sentence

L392-394: please revise

L397-399: in my opinion, this is a strong confirmation of a GWAS result by transcriptome analysis; it should be emphasized more

L430: “surprisingly” should be “surprising”

L457: “with deacclimation” should be removed

L461: “was” should be “were”

L465-468: The sentence would sound better as follows: “Thus, we might expect KS09068B-5-1 to show higher expression….”

L492: “there is “ should be “there are”

L520: is it really -20°C? or do you mean 20°C?

L552-553: did you use the whole SNP set for GWAS? Or did you filter for minor allele frequency and/or callrate? Markers showing low minor allele frequencies are not very helpful because their statistic significance is low.

L565: Haung should be Huang; “Wang et al. 2014” is missing in the reference list

L598: “TASSLE5” should be “TASSEL5”

L638: this reference is dated 1989 in the text

L652-655: references 16 and 17 are reported but not cited in the text

L658: ref. 18 is dated 2019 in the text

L664: ref. 20 is reported but not cited in the text

L691: ref. 30 is dated 2020 in the text

L716: a bracket is missing

Author Response

n this paper the authors aimed at identifying loci involved in the post-deacclimation tolerance to freezing in a population of Brassica napus accessions, using an association mapping approach. In parallel, on two varieties showing divergent tolerance to freezing, a transcriptome analysis was performed after an appropriate cold treatment to elucidate the molecular mechanisms underlying this trait. A number of markers associated to freezing tolerance were detected, and gene categories differentially expressed between the different treatments and varieties were described and discussed.

The work provides interesting information about the genetic determinants of tolerance traits as well as the physiological mechanisms involved; however, I have some concerns about the experimental workflow, which could be implemented, and in my opinion the paper suffers a few flaws that I suggest to address before publication.

 MAJOR CONCERNS:

  • L512-521: As far as I understood, the freeze treatment to the panel before GWAS was different from that to the two divergent varieties before transcriptome analysis. In my opinion this makes difficult to compare and cross-validate the results. Could you explain why two different treatments were used?

No good explanation except that science is sometimes messy. The initial screening for freezing tolerance following acclimation (being submitted in a different manuscript) was done to maximize the damage (an early evening freeze) by one of the collaborators. For consistence, the phenotyping for deacclimation GWAS was done the same way. For the assessment of the two chosen cultivars, it was decided to use a more physiologically relevant freezing regime where freezing was done in the morning with lights coming on during the last hour of the freezing and through the thawing process. The acclimation process was the same between the two studies (4 weeks at 5C). The deacclimation at 10C was chosen specifically since there appeared to be an odd difference between the response of the two cultivars at that temperature- note that there is not a lot of difference between them when fully acclimated or when deacclimation was at a higher temperature. This suggested there may be a difference in the deacclimation “switch” between these two varieties, and thus we decided to focus on that temperature for our deacclimation RNAseq analysis.

  • L345-350: the authors state that some mechanisms of post-deacclimation tolerance to freezing detected could be related to the basal (pre-acclimation) tolerance. Why not performing a GWAS on the same accessions for pre-acclimation tolerance? The two results could be compared and in this way possible overlapping could be evidenced. In my opinion this would implement the information about the mechanisms detected and would avoid to leave an open question.

This set of experiments are planned but have not yet been initiated. The manuscript is already quite long, and adding in an additional and potentially different GWAS would make this an even more unwieldy paper. Additionally, it would take the better part of a year to run the experiments given our greenhouse and freezing chamber space and the fairly large number of replicates we need to run because the freezing damage is not tightly uniform even within a given genotype. Thus, although we agree fully that this is an subject of interest, and that we are pursuing, I hope you can understand why it has not been included in this manuscript.

 GENERAL COMMENTS TO THE MANUSCRIPT

There are some grammar errors which need to be checked and fixed (e.g. L63: “although mutant analyses has implicated” should be “although mutant analyses HAVE implicated”)

Thanks. We have gone back over the manuscript and have tried to fix what we could find.

Some parts of the manuscript (especially those about the results of the trasncriptome analysis) are quite verbose and difficult to read. English should be revised to make the text more easily readable.

Again, we have gone back over the manuscript with an eye to reduce verbosity. I admit that is a failing of mine, but hopefully some of my co-authors have fixed some of these issues.

Please ensure that all the references are cited in the same way in the text (“et al” is sometimes followed by a dot and sometimes it’s not).

Hopefully we have found these and the copy editor is a good one if we have missed any.

In general, the Celsius degrees should be indicated with ”°C” and not ”C”; the Latin plant names should be written in italics (also in the references).

Fixed.

Below, more specific line by line comments:

L14-16: Affiliations 4 to 6 are not assigned to any of the authors

Fixed

L86: the reference cited are quite old. With a simple Pubmed search it’s possible to find more recent information about the identification of genes involved in cold tolerance. I suggest to add more up-to-date references

This was specifically looking at GWAS-based analyses in arabidopsis to be more relevant to that particular paragraph. I was surprised to see how few there were in the literature. I suspect this has to do with the difficulty in phenotyping. I did add in one of my colleague’s recent papers though on the subject. Hopefully this is sufficient to address your concern.

L104: the standard deviations of the phenotypic scores for frost damage are quite high; did you check for possible outliers?

We did not. This is not uncommon for visual scores of damage caused by freezing tolerance. I’ve done work in multiple brassica species, and there is a lot of stochasticity I the damage sustained. The source of this variation is not fully understood. We know that all of our plants freeze (based on some unpublished thermography studies we did), but manifestation of damage is sometime hit or miss- particularly in the intermediate temperatures where we can detect varietal differences. Thus, we have taken to running at least three replicates (and six in some related studies) with at least three technical replicates of each genotype in a random design. I certainly wish this were not reality or that we could figure out why some plants with the same genotype and under the same conditions die, and others survive. This was almost a theme at the meeting from which this special issue was solicited.  

L127 and 135: this is just an aesthetical observation, but I suggest to set fig. 2 and 3 at the same size

We have fixed this, though I suspect the copy editor will adjust it as well.

L208-220: this part is quite complicate to read; I think it should be simplified (especially L218-220)

Hopefully our changes have fixed this problem.

L250-253:  genes related to nitrogen, protein and RNA production are much more overexpressed during acclimation than during the following deacclimation, do you have any comments on this?

So these are all ontologies associated with protein production- from transcription through protein production and from nitrogen import to amino acid synthesis, modification, and transport. So, it is pretty big group. That said, it is easy to come up with a story for why these ontologies might be more prevalent during acclimation since there is a lot that has to happen when the plant is adjusting it’s physiology to cold. I was surprised not to see all of these groups reversing themselves as the plants deacclimated.  What this means is that these processes don’t change as much (in comparison to stress responses or photosynthesis and carbon utilization) when the acclimated plant deacclimates.

L261: “stress responses” should be removed

done

L266: the second comma should be removed

fixed

L277: generally speaking about this section, I think that it could be better readable if associated markers were separate in different paragraphs

This has now been done

L280: the reference to figure 8 should probably be to figure 10

Good catch!

L287: “maker” should be “marker”; if I understood correctly “genome” should be “gene”

Fixed (and the word search turned up one additional similar error. Thanks!)

L317: see L287 ^^

L324: in the manhattan plot, the title should be enlarged and fitted to the text; I suggest to include the names of the associated markers aside each arrow

This has been done – I didn’t realize U had clicked on something as I saved it from Tassel.

L327: in general, I suggest to write the name of each associated marker after citation

Hopefully the new figure and figure legend fix this issue.

L328-331: if 0.21 is the correlation coefficient (r2), it should be indicated and moved after “correlation”. “(as noted by Chao et al. 2020)” should be at the end of the sentence.

Hopefully the changes made make the point more clear.

L333-336: the verb is missing

Verb added.

L351-353: in the GWAS analysis, did you try a fixed significance threshold (es. 0.05) followed by a correction such as Bonferroni or a correction for multiple tests such as the Benjamini-Hochberg test for False Discovery Rate?

I must admit I am still a novice at this sort of analysis, but Tassel5 does employ a multiple testing parameter for the GLMs. This data is shown in supplemental file 9 on the designated page. However, in these more highly corrected models, In all of the analyses I have done, I have never seen an FDRs that fall below 0.1. I have read (see Estimation of a significance threshold for genome-wide association studies by Kaler and Purcel 2019) that these tend to be too stringent -particularly for more complex traits.  I have been told by our local expert that another way to select a threshold is to go with the expected probability of 0.01% of the markers. This comes out to just about 1E-4. Thus, the top 3 markers fall above this arbitrary threshold, and one just barely misses it. Our associations are admittedly weak, but we have presented all of the data (if one includes our paper on the marker and population characterization that should soon be published), so any reader with can make their own conclusions as to what is likely significant. We have some preliminary data that mutations in VIP3 and PSK4 in arabidopsis have some impact on deacclimation – and possibly intrinsic freezing tolerance, but we need to confirm our results. Thus, I don’t think we are way off base in our associated loci.

L374: the citation “Leene et al 2019” should be “van Leene et al. 2019”

Good catch

L380-381: please revise this sentence

Hopefully the rephrasing makes it more clear.Thus, it may be probable that TOR-regulated growth inhibition could provide a possible mechanism for controlling variation in freezing survival following deacclimation.”

L392-394: please revise

This sentence now reads: Feng et al. 2017 has proposed a that the vernalization response reduces the ability of the plant to maintain freezing tolerance.

L397-399: in my opinion, this is a strong confirmation of a GWAS result by transcriptome analysis; it should be emphasized more.

I would love to agree with you, but there are quite a few genes that are differentially regulated between acclimated and deacclimated plants. It could well be chance. However, I have highlighted that gene in the new table requested by reviewer one. And have strengthened the comment in the discussion.

L430: “surprisingly” should be “surprising”

Fixed

L457: “with deacclimation” should be removed

Fixed

L461: “was” should be “were”

Fixed

L465-468: The sentence would sound better as follows: “Thus, we might expect KS09068B-5-1 to show higher expression….”

Adjusted the sentence accordingly.

L492: “there is “ should be “there are”

Fixed

L520: is it really -20°C? or do you mean 20°C?

Fixed

L552-553: did you use the whole SNP set for GWAS? Or did you filter for minor allele frequency and/or callrate? Markers showing low minor allele frequencies are not very helpful because their statistic significance is low.

I understand that statistical significance is low for SNPS with low MAFs. However, it was previously suggested to me to leave them in since they may still actually be significant even though their power is low. I have run the analyses with a MAF cut off at >0.05%. It reduced the number of SNPs by a little over half. All of the significant snps have more than 3 minor alleles with most over 10 present in the population with half (now highlighted in yellow in supplemental file 9) having a MAF >0.05.

L565: Haung should be Huang; “Wang et al. 2014” is missing in the reference list

Fixed

L598: “TASSLE5” should be “TASSEL5”

Fixed

L638: this reference is dated 1989 in the text

Fixed

L652-655: references 16 and 17 are reported but not cited in the text

Fixed

L658: ref. 18 is dated 2019 in the text

Fixed

L664: ref. 20 is reported but not cited in the text

Fixed

L691: ref. 30 is dated 2020 in the text

Fixed

L716: a bracket is missing

Fixed

Round 2

Reviewer 4 Report

The comments and observations on the paper have been mostly adequately addressed, and all the points have been fixed/justified. However, I still detected  a number of flaws which should be corrected, listed below.

____

In the title, Brassica napus should be written in full (not “B. napus”) as it’s the first citation, and should be in italics.

L215-233: in my opinion this part is generally still too complicated. References to the specific sheets of supplementary files were added, but the text should be simplified as well. I suggest some specific modifications below.

L222-224: if I have correctly understood the meaning, it might better sound as: “This supported the over-representation of a large number of ERF/AP2 transcription factor regulated genes detected in ChIP studies”.

L226-228: This sentence could be rephrased as: “This also compliments the observation from the clusters of significantly over-represented GO ontologies among genes that were either up or down regulated during acclimation in both varieties”.

L345: If I understand correctly this sentence in the legend of fig. 10 needs revision. Please check

L420: please check also this sentence (“has proposed a that the vernalization…..”)

Author Response

In the title, Brassica napus should be written in full (not “B. napus”) as it’s the first citation, and should be in italics.

Fixed

L215-233: in my opinion this part is generally still too complicated. References to the specific sheets of supplementary files were added, but the text should be simplified as well. I suggest some specific modifications below.

Thank you for taking the time to make the suggestions. I had two other authors go back over it to try to simplify it, but much of this is style as of anything else. I am happy to make any suggested alterations that don’t alter the meaning of the sentences.

L222-224: if I have correctly understood the meaning, it might better sound as: “This supported the over-representation of a large number of ERF/AP2 transcription factor regulated genes detected in ChIP studies”.

Fixed

L226-228: This sentence could be rephrased as: “This also compliments the observation from the clusters of significantly over-represented GO ontologies among genes that were either up or down regulated during acclimation in both varieties”.

Fixed

L345: If I understand correctly this sentence in the legend of fig. 10 needs revision. Please check

Fixed (extra “was” removed)

L420: please check also this sentence (“has proposed a that the vernalization…..”)

Excellent eyes!  Fixed.

Thanks again for the very complete and helpful review!

This manuscript is a resubmission of an earlier submission. The following is a list of the peer review reports and author responses from that submission.

Round 1

Reviewer 1 Report

Excellent paper.

Very minor editorial suggestions:

L58. Plural issue in sentence structure

L123. should be "days" not "day"

L302 remove hyphen

303 The term "destroyed" is a bit dramatic.  Suggest "killed"

L316. Confusing sentence,  Please re-word

L327 "observed to be" not necessary.  Suggest you remove.

L389 "as" instead of "a"

L398 "windy" inside a greenhouse?  Poor choice of word.

L438 to L445 Are light levels known?  Not a deal breaker but would be nice to know.

Author Response

Very minor editorial suggestions:

L58. Plural issue in sentence structure Fixed

L123. should be "days" not "day" Fixed

L302 remove hyphen Fixed

303 The term "destroyed" is a bit dramatic.  Suggest "killed" Fixed

L316. Confusing sentence,  Please re-word Now reads “Establishing parameters for significance of association between the phenotype and genotype is somewhat arbitrary. In this study, the top four were chosen (p < 3.97E-05 which is the significant value for the top 0.004% of the markers).”

L327 "observed to be" not necessary.  Suggest you remove. Fixed

L389 "as" instead of "a" Fixed

L398 "windy" inside a greenhouse?  Poor choice of word. The cooling system has fans which are rather strong.

L438 to L445 Are light levels known?  Not a deal breaker but would be nice to know.

We have not measured the light levels, but I have included the model and make of the lights we use in the chamber in the methods section to provide a rough estimate.

Reviewer 2 Report

              This manuscript “Genome wide association studies and transcriptome changes during acclimation and deacclimation in divergent canola varieties”is dealing with huge datasets regarding two canola varieties that show different response to deacclimation treatment. From 397 canola varieties, the authors selected two canola varieties showing different survival scores after deacclimation treatment at 10°C, but similar scores after cold acclimation. Since these two varieties are expected to have different deacclimation mechanisms, RNAseq analysis was performed to identify differentially expressed genes between these two varieties. In general, deacclimation had opposite responses to cold acclimation. However, a number of genes associated with oxidative stress and photosynthesis were identified with differential deacclimation responses between two canola varieties and deduced to be crucial for regulation of freezing tolerance after deacclimation. In addition, GWAS studies identified polymorphisms at several different loci which is expected to be associated with freezing tolerance at deacclimation stages.

              This study provides many insights on not only future advances in improvements of cold tolerance in crops, but also physiological mechanisms of a number of genes associated with loss of plant freezing tolerance in response to deacclimation treatment. Since understanding of the mechanisms of deacclimation is very limited so far, interpretation of the RNAseq and GWAS datasets obtained in the present study could be a foundation for further investigations. However, there are several points where I think this paper needs to be addressed and clarified.

  1. Please explain justifiable reasons why canola was chosen as a plant material in present study. Importance of experimental materials from the aspect of scientific and/or agricultural values need to be described in the introduction.
  2. How about basal freezing tolerance of two canola varieties (KS4666 and KS09068B-5-1)?
  3. 2: Treatment at 5°C after 4 weeks cold acclimation should not be “deacclimation”. This treatment is simply named as e.g. “extended cold acclimation” otherwise readers will misunderstand the results.
  4. The colors, fonts, and formats of the figures are inconsistent, especially in Fig. 2 and 3. These figures don’t have labels for vertical axis.
  5. Although authors are mentioned about growth rate and vernalization at deacclimation stage in discussion (L346, L356), are there any different changes in appearance between the two varieties during acclimation and deacclimation? Isn’t it necessary to consider changes in the growth stage during or after deacclimation process? It can be expected that growth will almost completely stop or be significantly delayed in the cold acclimation process, but in such a deacclimation environment, it seems that the expression of many genes can be altered due to the progress of growth, vernalization and/or senescence. Therefore, I think it is better to be careful about simply comparing the gene expression patterns in cold acclimation and deacclimation and linking them directly to the difference in freeze tolerance.
  6. L151: “A core set of 675 genes” should be rephrased more specifically.
  7. 5 simply focuses on genes that show opposite responsiveness in acclimation and deacclimation, but there may be important genes that continuously increase or decrease in the process from acclimation to deacclimation (for example, genes involved in low temperature memory). Did you find any remarkable genes in the above comparison?
  8. L196-L202: Are these results shown in figures? Supplemental table 2 should be referred here.
  9. It is necessary to explain the criteria by which the motifs listed in Figure 6 and 7 were picked up and sorted (probably sorted by lowest P-value?).
  10. L417: “Deacclimation generally had the opposite response to acclimation” Are there any relevant studies previously demonstrated or suggested in other plant species such as Arabidopsis thaliana regarding this statement? I think several studies mentioned opposite responses in cold acclimation and deacclimation. The similarities regarding such responses with other plants should be discussed.
  11. All gene names should be italicized. In addition, there are many various formatting mistakes and typos in the manuscript. It is necessary to carefully proofread the manuscript again.

Author Response

  1. Please explain justifiable reasons why canola was chosen as a plant material in present study. Importance of experimental materials from the aspect of scientific and/or agricultural values need to be described in the introduction.

Several sentences describing the agronomic uses of canola were added to the beginning of the introduction.

  1. How about basal freezing tolerance of two canola varieties (KS4666 and KS09068B-5-1)?

This is a good question- particularly since survival after deacclimation could be due to high basal (unacclimated) level of freezing tolerance. We did present data that indicated unacclimated plants were killed by a -10 freeze, but clearly some lines survive freezing following acclimation and deacclimation. Thus, we expect some of the survival to result from lack of complete deacclimation. However, we have not measured basal levels of freezing tolerance in our diversity panel at temperatures other than -10C. We have added a couple of sentences to the end of the first paragraph of the discussion to note this possibility.

  1. 2: Treatment at 5°C after 4 weeks cold acclimation should not be “deacclimation”. This treatment is simply named as e.g. “extended cold acclimation” otherwise readers will misunderstand the results.

Figure legend now reads “Average freezing damage scores for two divergent canola varieties (KS4666 and KS09068B-5-1) in fully acclimated plants 5C, or following deacclimation at 10C, or 15C for three days after a 4-week acclimation period (5C).”

  1. The colors, fonts, and formats of the figures are inconsistent, especially in Fig. 2 and 3. These figures don’t have labels for vertical axis.

The description of the Y axis is now noted in the figure legend. The colors do not need to match since figures 2 and 3 they are describing different data.

  1. Although authors are mentioned about growth rate and vernalization at deacclimation stage in discussion (L346, L356), are there any different changes in appearance between the two varieties during acclimation and deacclimation? Isn’t it necessary to consider changes in the growth stage during or after deacclimation process? It can be expected that growth will almost completely stop or be significantly delayed in the cold acclimation process, but in such a deacclimation environment, it seems that the expression of many genes can be altered due to the progress of growth, vernalization and/or senescence. Therefore, I think it is better to be careful about simply comparing the gene expression patterns in cold acclimation and deacclimation and linking them directly to the difference in freeze tolerance.

We did not notice any differences in growth between the two varieties, although we did not specifically measure growth during the cold acclimation process. We didn’t consider vernalization to be a significant issue since both lines are winter varieties and thus likely both undergoing the early stages of vernalization. We generally do not see vernalization of our winter lines prior to 6 weeks cold, although we have not specifically looked at differences in vernalization in our diversity panel. However, since we have not determined the vernalization status of these two varieties after 4 weeks, it is possible that some of the changes in gene expression between the lines could be due to faster or slower vernalization processes in addition to acclimation processes. We have added some lines to the discussion to note this possibility. We have added the following to the discussion: It should be noted that we have not assessed the timing of vernalization processes attributed to either of our tested varieties. Although neither variety showed obvious differences in growth or flowering during the deacclimation process, it is possible that some of the differences in gene expression and freezing tolerance could be due in part to potential differences in the developmental state of either variety. Further study will be needed to determine the role – if any- of flowering capacity following 4 week acclimation processes.  

  1. L151: “A core set of 675 genes” should be rephrased more specifically.

The sentence now reads “A set of 675 genes were differentially expressed between these two varieties under all conditions.” Hopefully that solve the problem.

  1. 5 simply focuses on genes that show opposite responsiveness in acclimation and deacclimation, but there may be important genes that continuously increase or decrease in the process from acclimation to deacclimation (for example, genes involved in low temperature memory). Did you find any remarkable genes in the above comparison?

There were certainly genes that were differential from both acclimated vs control and deacclimated vs control (in the same direction for both). However, it would be difficult to ascertain if those differences related to “memory” per se since one of the varieties did not seem to show significant deacclimation. The expression data is available for anyone who wants to look specifically at those genes in the  variety KS09068B-5-1. But one should remember that although freezing tolerance was largely lost in that variety, the plants were kept at 10C which could be maintaining some acclimating responses. That is an interesting question though.

  1. L196-L202: Are these results shown in figures? Supplemental table 2 should be referred here.

That data comes from supplemental file 1. I now reference that information in that sentence.

  1. It is necessary to explain the criteria by which the motifs listed in Figure 6 and 7 were picked up and sorted (probably sorted by lowest P-value?).

Yes, and this is now noted in the figure legend.

  1. L417: “Deacclimation generally had the opposite response to acclimation” Are there any relevant studies previously demonstrated or suggested in other plant species such as Arabidopsis thaliana regarding this statement? I think several studies mentioned opposite responses in cold acclimation and deacclimation. The similarities regarding such responses with other plants should be discussed.

Yes, and this was also pointed out as a constructive criticism by reviewer 3. Additional references have been added to the discussion.

  1. All gene names should be italicized. In addition, there are many various formatting mistakes and typos in the manuscript. It is necessary to carefully proofread the manuscript again.

The manuscript was reviewed again for such errors.

Reviewer 3 Report

The MS attempted to use GWAS to identify genomic loci and genes associated with deacclimation in 397 winter canola varieties and RNAseq analysis to identify transcriptome changes in two extreme varieties. Deacclimation resistance is important during the transition from winter to spring and is a major determinant of plant hardiness. As such, a study of deacclimation was warranted. Unfortunately, the authors did not exploit the huge data set available to them in the GWAS study. Similarly, the RNAseq analysis was poorly presented. The MS lacks direction and seems to have been hastily put together.

Major revisions:

GWAS:

  1. In the GWAS the authors did not mention the type and number of markers that were used to genotype the 397 winter canola varieties.
  2. They did not explore population structure of the 397 diversity set (this should be summarized even if this is contained in the submitted MS- Horvath et al. 2020).
  3. The authors did not provide any data (QQ or Manhattan plots) for the GLM and MLM models. What was the point in carrying out these models under PCA of 1 and 17? The standard is for the first three PCA.
  4. In addition, did the FARMCPU, BLINK and SUPER software yield better MTAs than the GLM and MLM models?
  5. The data of the damage scores cannot be inferred from Figure 1. This should be presented as a supplementary table. In addition, the preliminary experiment data is not presented.
  6. Better presentation of the phenotypic data will justify why the authors selected KS4666 and KS09068B-5-1 for the RNAseq.

RNA-seq:

  1. The RNAseq analysis identified a number of differentially expressed genes. This does not mean much without a proposed model on the interaction of the genes during deacclimation.
  2. The genome of Brassica rapa and B. napus are fully annotated. Hence, the use of only Arabidopsis thaliana genome (Figures 6 and 7) to identify gene function is not thorough enough.
  3. The authors did not use of heat map to illustrate the regulation of key genes.

Minor revisions

  1. Line 41: Brassica napus should be italicized “Brassica napus
  2. Line 82: deacclimation
  3. Line 83: This study “examined” varietal differences------
  4. Line 92: Number of varieties examined should be stated
  5. Line 99-101: The authors should state exact numbers or percentages and not resort to the use of ‘some lines’ and ‘other lines’.
  6. Figure 2 and 3 legends should explain the data and not just rating scale used.
  7. Line 205-206: “Likewise,-----transcription factors.” should be moved to the “Discussion”
  8. Line 213-214: “Genes -------well studied.” Should be moved to the “Discussion”
  9. Line 236: both varieties? KS4666 and KS09068B-5-1 or KS09068B-5-1 and KS4666
  10. Line 241: Did the authors observe the down-regulation of ABA, SA and JA? To what extent?
  11. Line 259: Position 37666286 on chromosome Ann_random? This is not clear.
  12. Line 261-264: did the authors perform linkage decay on their data set or was inferred from Horvath et al. (2020).
  13. The discussion is poorly written and generally failed to compare the results of this study to published literature.

Author Response

Major revisions:

GWAS:

  1. In the GWAS the authors did not mention the type and number of markers that were used to genotype the 397 winter canola varieties.

We have added that information to the Methods

  1. They did not explore population structure of the 397 diversity set (this should be summarized even if this is contained in the submitted MS- Horvath et al. 2020).

We have added that information to the Methods

  1. The authors did not provide any data (QQ or Manhattan plots) for the GLM and MLM models. What was the point in carrying out these models under PCA of 1 and 17? The standard is for the first three PCA.

WOW!  That is embarrassing for a lot of people (mostly me). I cannot believe I forgot to include the Tassel output file, and that none of my colleagues realized I didn’t nor any f the other two reviewers. Thank you for being the sharp one!!! I have now included the Tassel output table along with the MSD values to identify the best model fit as a supplemental file. I have qq plots from all models and they are all fairly linear once one includes the principle components into the model although the most linear is with a GLM model with a PC of 17. Best practice, as I understand it is use PCA 1 and a PCA that covers 25% of the diversity (that was obtained with 17 PCs). This additional analysis also pointed to the identification of one additional loci on Chrm A06. So again, thank you for catching this! We also now include a Manhatten plot that best highlights the now 6 best associated loci.

  1. In addition, did the FARMCPU, BLINK and SUPER software yield better MTAs than the GLM and MLM models?

Slightly better, but mostly we were looking for overlap between models. The consensus I am getting from talking to people about this is that one should run multiple models and then look for markers that are near the top in most of them. In this case, although the marker was in the top 500 of the MLM with kinship and a PC of 17, when the p values of all of the models were averaged, this marker was in the top 50.

  1. The data of the damage scores cannot be inferred from Figure 1. This should be presented as a supplementary table. In addition, the preliminary experiment data is not presented.

We agree and have added the data used for that graph as a supplemental file (supplemental file 1).

  1. Better presentation of the phenotypic data will justify why the authors selected KS4666 and KS09068B-5-1 for the RNAseq.

Hopefully the data added in the new supplemental file will suffice. Incidentally, those lines were chosen based on earlier data in which variety KS4666was among the top surviving varieties. However, after subsequent experimental runs, it fell in the middle of the pack. Thus, the lack of significant loss of deacclimation on this line was fortuitous. Indeed, the same damage score (of 1) was observed in a set of 9 replicated runs on fully acclimated plants (although the freezing regime was more intense for those studies).

RNA-seq:

  1. The RNAseq analysis identified a number of differentially expressed genes. This does not mean much without a proposed model on the interaction of the genes during deacclimation.

We agree, but there really isn’t a model to propose based on this data. We did not find any obvious regulatory genes associated with the deacclimation process as was shown by the GSEA data. Thus, we do can not predict a model based on our data.

  1. The genome of Brassica rapa and B. napus are fully annotated. Hence, the use of only Arabidopsis thaliana genome (Figures 6 and 7) to identify gene function is not thorough enough.

We agree somewhat: The annotations of the best arabidopsis hit are more informative than the Bnn gene model annotations. However, we now include both in the supplemental file 2 (previously supplemental file 1). The arabidopsis orthologues were used for the GSEA since the program only accepts arabidopsis AGI annotation.

  1. The authors did not use of heat map to illustrate the regulation of key genes.

We are not great fans of heatmaps. They may be pretty, but too often lack the definition needed to derive any but the most raw indications of the proportion of up- or down-regulated genes (data provided by the numbers presented in the results). We have opted to provide instead the rich and detailed data in what is now supplemental file 2. That data has all of the averaged expression data along with FDR values and is very well annotated by function. Any interested party can easily sort the data by any criteria to pull information they are interested in.

Minor revisions

  1. Line 41: Brassica napus should be italicized “Brassica napus” Fixed
  2. Line 82: deacclimation Fixed
  3. Line 83: This study “examined” varietal differences------b Fixed
  4. Line 92: Number of varieties examined should be stated Fixed
  5. Line 99-101: The authors should state exact numbers or percentages and not resort to the use of ‘some lines’ and ‘other lines’. Fixed
  6. Figure 2 and 3 legends should explain the data and not just rating scale used. Fixed (I hope).
  7. Line 205-206: “Likewise,-----transcription factors.” should be moved to the “Discussion”

No, this actually refers to our data and I now again refer it to supplemental file 3.

  1. Line 213-214: “Genes -------well studied.” Should be moved to the “Discussion”

Similar to above. This is our data and as refered to in the following sentence.

  1. Line 236: both varieties? KS4666 and KS09068B-5-1 or KS09068B-5-1 and KS4666

The “respectively refers to the up vs down-regulated genes. These are ones that overlap in both varieties.

  1. Line 241: Did the authors observe the down-regulation of ABA, SA and JA? To what extent?

This is gene set enrichment data. The actual genes in each association set are listed in the supplemental files.

  1. Line 259: Position 37666286 on chromosome Ann_random? This is not clear.

Not sure what is wrong here. The French assembly contains sets of contigs associated with some chromosomes or some genomes. This position is present in the genome browser.

  1. Line 261-264: did the authors perform linkage decay on their data set or was inferred from Horvath et al. (2020).
  2. The discussion is poorly written and generally failed to compare the results of this study to published literature.

Linkage decay will be the same for this population as it was determined from all available markers from all available varieties in the population.

Round 2

Reviewer 3 Report

I still find the MS to be heavy on data but these were poorly presented. Readers are required to shift through numerous files to infer and draw conclusions.  The authors should summarize the important findings in all the supplementary files and include them in the main MS. In addition, there was the over-reliance on Arabidopsis thaliana genome instead of B. napus genome (multiple assemblies are available at NCBI) to identify gene function. I recommend re-analysis of the data and major revisions to address these concerns.